# The Clinical and Genetic Characteristics of *Streptococcus agalactiae* Meningitis in Neonates

**DOI:** 10.3390/ijms242015387

**Published:** 2023-10-20

**Authors:** Jen-Fu Hsu, Jang-Jih Lu, Shih-Ming Chu, Wei-Ju Lee, Hsuan-Rong Huang, Ming-Chou Chiang, Peng-Hong Yang, Ming-Horng Tsai

**Affiliations:** 1Division of Pediatric Neonatology, Department of Pediatrics, Chang Gung Memorial Hospital, Taoyuan 333, Taiwan; jeff0724@gmail.com (J.-F.H.); kz6479@cgmh.org.tw (S.-M.C.); weijulee@cgmh.org.tw (W.-J.L.); qbonbon@gmail.com (H.-R.H.); cmc123@cgmh.org.tw (M.-C.C.); ph6619@cgmh.org.tw (P.-H.Y.); 2School of Medicine, College of Medicine, Chang Gung University, Taoyuan 333, Taiwan; janglu45@gmail.com; 3Department of Laboratory Medicine, Linkou Chang Gung Memorial Hospital, Taoyuan 333, Taiwan; 4Department of Medical Biotechnology and Laboratory Science, Chang Gung University, Taoyuan 333, Taiwan; 5Division of Neonatology and Pediatric Hematology-Oncology, Department of Pediatrics, Chang Gung Memorial Hospital, Yunlin 638, Taiwan

**Keywords:** Group B *Streptococcus*, serotype III/CC17 GBS, multilocus sequence typing, antimicrobial resistance, neonatal meningitis

## Abstract

*Streptococcus agalactiae* (Group B *Streptococcus*, GBS) is an important pathogen of bacterial meningitis in neonates. We aimed to investigate the clinical and genetic characteristics of neonatal GBS meningitis. All neonates with GBS meningitis at a tertiary level medical center in Taiwan between 2003 and 2020 were analyzed. Capsule serotyping, multilocus sequence typing, antimicrobial resistance, and whole-genome sequencing (WGS) were performed on the GBS isolates. We identified 48 neonates with GBS meningitis and 140 neonates with GBS sepsis. Neonates with GBS meningitis had significantly more severe clinical symptoms; thirty-seven neonates (77.8%) had neurological complications; seven (14.6%) neonates died; and 17 (41.5%) survivors had neurological sequelae at discharge. The most common serotypes that caused meningitis in neonates were type III (68.8%), Ia (20.8%), and Ib (8.3%). Sequence type (ST) is highly correlated with serotypes, and ST17/III GBS accounted for more than half of GBS meningitis cases (56.3%, *n* = 27), followed by ST19/Ia, ST23/Ia, and ST12/Ib. All GBS isolates were sensitive to ampicillin, but a high resistance rates of 72.3% and 70.7% to erythromycin and clindamycin, respectively, were noted in the cohort. The virulence and pilus genes varied greatly between different GBS serotypes. WGS analyses showed that the presence of PezT; BspC; and ICE*Sag37* was likely associated with the occurrence of meningitis and was documented in 60.4%, 77.1%, and 52.1% of the GBS isolates that caused neonatal meningitis. We concluded that GBS meningitis can cause serious morbidity in neonates. Further experimental models are warranted to investigate the clinical and genetic relevance of GBS meningitis. Specific GBS strains that likely cause meningitis requires further investigation and clinical attention.

## 1. Introduction

*Streptococcus agalactiae* (Group B *Streptococcus*, GBS) is one of the most common pathogens causing neonatal sepsis and meningitis [1,2,3]. GBS sepsis in neonates can be categorized as early-onset disease (EOD, disease onset before the 7th day of life) and late-onset disease (LOD, disease occurring after the 8th day of life), where both are associated with a high mortality rate of 8–14% and a high risk of neurological complications, especially in neonates with GBS meningitis [4,5,6]. Although the policies of intrapartum antibiotic prophylaxis and routine maternal screening have decreased the incidence of GBS EOD, GBS LOD remains an important threat that causes mortality in the neonatal intensive care unit (NICU) [6,7,8]. Recently, researchers have focused on the molecular epidemiology, predominant type III GBS sepsis in neonates, and genetic characteristics of specific GBS isolates that are associated with GBS sepsis and meningitis [9,10,11,12,13].

GBS meningitis is associated with a high mortality rate and long-term neurological disability in neonates [6,10,14,15]. The high rates of neuropsychiatric and developmental sequelae among GBS meningitis survivors highlight the importance of understanding the pathophysiology and genetic mechanisms, which may help clinicians to develop new therapeutic and/or neuroprotective strategies for the developing brain [15,16,17]. The serotype III GBS hypervirulent clonal complex 17 (CC17) isolates have become a predominant cause of meningitis and GBS LOD in recent years, which may be related to their higher penetration ability to cross the blood–brain barrier (BBB) [18,19,20]. However, the relevant molecular and genetic characteristics of specific GBS strains that are likely to cause meningitis remain unidentified. In this study, we aimed to investigate the clinical and genetic characteristics of GBS isolates that cause neonatal meningitis.

## 2. Results

### 2.1. Clinical Characteristics of Neonates with GBS Meningitis

During the study period, a total of 48 neonates with documented GBS meningitis and 140 cases of neonatal GBS sepsis without meningitis were identified and enrolled for analyses. The median (interquartile range [IQR]) gestational age and birth body weight (BBW) of the neonates with GBS meningitis were 38.0 (37.0–39.0) weeks and 2900 (2651.3–3251.3) g, respectively. The majority of cases were term-born neonates, and female patients outnumbered male patients. The patients’ demographics, clinical manifestations, and laboratory results are summarized in Table 1. Neonates with GBS meningitis were not significantly different from those with GBS sepsis without meningitis in patients’ characteristics, but they had significantly more severe symptoms, including a greater likelihood of having respiratory distress, severe sepsis, metabolic acidosis, and coagulopathy (Table 1). Of note, 25% of cases with GBS meningitis were EOD, and 27.3% of the patients with GBS EOD had meningitis.

All cases of invasive GBS diseases were treated with ampicillin. However, neonates with GBS meningitis were treated for a significantly longer duration than those with GBS sepsis (median 22.5 days vs. 12.5 days, *p* < 0.001). We found that neonates with GBS meningitis had a significantly higher percentage of neurological complications (n = 37, 77.8%) and the survivors were more likely to have neurological sequelae (n = 17, 41.5%) at discharge than the neonates with GBS sepsis (both *p* < 0.001) (Table 2). The mortality rate of the neonates with GBS meningitis was 14.6% (a total of seven neonates died), which was also higher than that of the neonates with GBS sepsis (Table 2).

### 2.2. Molecular Characteristics and Antimicrobial Resistance of GBS Isolates That Caused Meningitis

The molecular epidemiology and genetic characteristics of the GBS isolates that caused meningitis in our cohort are summarized in Table 3. Type III GBS accounted for 68.8% (n = 33) of all meningitis cases, followed by serotype Ia (20.8%, n = 10), Ib (8.3%, n = 4), and type II (2.1%, n = 1). Sequence type (ST) is highly correlated with serotypes, and ST17/III GBS accounted for more than half of GBS meningitis cases (56.3%, n = 27), followed by ST19/Ia (10.4%, n = 5) and ST23/Ia or Ib (10.4%, n = 5). Of note, the CC17 GBS isolates (n = 27, 56.3%) and CC12 (n = 6, 12.5%) most commonly caused neonatal meningitis in this study.

All the GBS isolates of neonates with meningitis and those from neonatal sepsis were susceptible to penicillin, ampicillin, vancomycin, and cefotaxime. High resistance rates of 72.3% and 70.7% to erythromycin and clindamycin, respectively, were noted in all GBS isolates. The antibiotic resistance profiles between the GBS isolates of neonatal meningitis and those of neonatal sepsis without meningitis were not significantly different, but a specifically high antibiotic resistance rate was noted in type III (77.1–82.2%), type Ib (100%), and type V (85.7%) GBS isolates. Due to the high correlation between the serotype and sequence type, the antibiotic resistance rate to erythromycin and clindamycin was also high in the CC12 (100%) and CC17 (89.3%) GBS isolates. Additionally, most of the GBS isolates (94.9%) with resistance to erythromycin were also resistant to clindamycin.

### 2.3. WGS for GBS Isolates That Caused Neonatal Meningitis

WGS was performed using three type III/ST17 GBS strains, one type Ib/ST12, and three type VI/ST-1 GBS strains, representing GBS meningitis and GBS sepsis without meningitis, respectively. Strain CP012480.1, which is also a type III/ST17 GBS isolate from neonates with invasive GBS diseases, was used as the reference strain. One V/ST1 GBS reference strain (CP010867) from NCBI was also used as the reference strain for comparison. Comparative genome analyses were performed for all GBS isolates and the two reference strains to track the possible genomics of mobile elements and insertion sequences (IS). All genes related to the component systems CovS/R, antibiotic resistance, pilus formation, capsular serotype, and virulence were investigated.

Concerning the genes of component systems CovS/R, most antimicrobial resistance genes and most virulence genes were not significantly different between the GBS strains of neonatal meningitis and those of neonatal sepsis without meningitis (Figure 1). However, the presence of BspC, ICE*Sag37*, and PezT was noted in GBS isolates of neonatal meningitis, both in type Ib/CC12 and III/CC17 GBS, but not in the type VI/CC1 GBS strain. The BspC, PezT, and HvgA genes, as well as the pili, are previously reported to be associated with enhanced virulence and pathogenic mechanisms, which lead to life-threatening illness [15,19]. Additionally, the PezT, ICE*Sag37*, and BspC genes were not found in the reference CP012480.1 strain (also type III/ST strain from neonatal sepsis). Therefore, we highly suspected that the BspC, PezT, and ICE*Sag37* genes were associated with the occurrence of meningitis in the neonates examined in this study.

PCR was performed for all type Ib and type III GBS isolates to verify the results and confirm the presence of multiple genes in all clinical GBS isolates. The primers used for all targeted genes, including BspC, ICE*Sag37*, PezT, and HvgA are summarized in Table 4. The genetic characteristics of all GBS isolates are listed in Table 3. We found the presence of PezT, BspC, and ICE*Sag37* in 60.4%, 77.1%, and 52.1% of all GBS isolates that caused neonatal meningitis in our cohort. Additionally, HvgA was present in 80% of the type III/CC17 GBS isolates of neonatal meningitis.

### 2.4. Discussion

Compared to numerous studies that have investigated the molecular epidemiology of neonatal GBS invasive diseases and colonization in pregnant women, relatively fewer studies have focused on GBS meningitis [4,21,22]. We found that GBS meningitis is associated with more severe clinical manifestations and worse long-term neurological outcomes in neonates. Most of the GBS isolates that caused neonatal meningitis in our cohort belong to the type III/CC17 and type Ib/CC12 strains, which is compatible with previous studies [4,6,21,22,23,24]. WGS was used to investigate the genetic differences between the GBS isolates of neonatal meningitis and those that caused neonatal sepsis in our cohort, and we found the presence of BspC, HvgA, and PezT genes, as well as ICE*Sag37*, to be potentially associated with the occurrence of GBS meningitis. We highlight the requirement of further in vivo and in vitro studies to document the roles and underlying mechanisms of these genes, which can be the future bases of preventive and therapeutic strategies.

In our institute, we have applied the policies of routine GBS screening for pregnant women and intrapartum antibiotic prophylaxis (IAP) since 2004. A change in policy was associated with the predominance of serotype III/ST-17 GBS isolates since 2010 and the increasing prevalence of antibiotic-resistant GBS isolates [12,13,20,25]. More infection control strategies have been carried out in Taiwan in recent decades, which has ultimately led to decreased cases of GBS sepsis and meningitis in neonates since 2018 [6,26,27,28]. However, in neonates with GBS infections, the chance of progression to severe sepsis and meningitis remains unchanged and the mechanisms are not yet fully understood. Therefore, we sought to investigate the genetic characteristics of GBS isolates associated with the occurrence of meningitis.

Various virulence genes and mechanisms have been documented to be associated with the occurrence of GBS meningitis in neonates, such as the enhanced penetration of the BBB caused by HvgA genes, pili, and Srr1/Srr2 genes [15,22,29,30,31]. In addition to the well-known type III GBS specific HvgA genes, we found BspC and PezT genes in most GBS isolates of neonatal meningitis. The BspC gene has been found to interact with host vimentin to promote bacterial adherence to the brain endothelium and inflammation in the in vivo and in vitro models of meningitis, which also leads to the enhanced penetration of the BBB and the occurrence of meningitis [32]. Therefore, targeting the BspC–vimentin interaction is a potential therapeutic strategy to decrease inflammation during GBS meningitis [33]. Although not all GBS isolates that caused meningitis carried both BspC and PezT genes, we suspected that the occurrence of GBS meningitis is associated with both host factors and the virulence of highly pathogenic GBS strains [23,34,35].

Pili are known to be associated with enhanced cell adhesion of GBS isolates, which contributes to transcytosis of the endothelium and increased penetration of the BBB. Therefore, the specific pilus profiles are reported to be associated with sepsis and meningitis [30,31]. Previous studies have concluded that specific pilus profiles in some GBS strains or phylogenetic lineages are associated with a higher risk of meningitis [31,36,37]. While the presence of PI-1 and PI-2b genes is noted in most of the type III/CC17 GBS isolates [36,37], our data showed that the presence of ICE*Sag37*, carrying multiple virulence genes and replacing the PI-1, is correlated with the clonal expansion of antibiotic-resistant type III/CC17 GBS strains, which may be associated with the occurrence of meningitis [19].

The PezT gene was initially found in *Streptococcus pneumonia* and is part of the epsilon/zeta system (the PezA/T system), which is a special toxin/antitoxin system that protects the bacterium itself [38,39]. PezT can activate the release of specific toxins to attack host cells or competing microorganisms, which has been considered a pathogenic and invasive mechanism [38,39]. The PezT affects the cell wall integrity via phosphorylase activities and increases the GBS invasiveness after epithelial lysis, which may contribute to the penetration of the BBB [38,39]. In contrast to a previous study that documented that the PezA/T system is unique in the CC12 GBS strain and contributes to higher virulence [40], we detected the PezT gene in 60.4% of all GBS isolates of neonatal meningitis. The PezA/T system is associated with a higher severity of illness in both the CC12 GBS strain and some non-CC12 GBS strains and deserves further study to investigate its involvement in neonatal meningitis.

An obviously decreased GBS sepsis-attributable mortality rate has been reported in recent decades, from more than 20% in the 1980s to approximately 10.7% in recent studies [3,4,5,41,42]. Improved critical care in neonates and the early implementation of prophylactic antibiotics may account for the reduced mortality rate [21,24]. However, a high rate of neurological complications and long-term neurological sequelae in survivors of GBS sepsis is currently the major concern. In our cohort, GBS EOD was related to a higher percentage of meningitis (27.3%) than other studies, which reported that most GBS meningitis cases are LOD. Additionally, GBS EOD was associated with a significantly higher rate of severe sepsis and a higher mortality rate than GBS LOD. Therefore, further studies for the early identification of specific strains, risk factors, or genetic mechanisms that cause GBS meningitis are urgently needed in the future.

Some limitations in this study should be addressed. All the invasive GBS isolates in this study were from a single center in Taiwan over an extensive period of time. During the prolonged timeframe, the therapeutic and IAP policies may have been changed. Therefore, a multicenter study is needed to investigate the epidemiological and genetic trends of GBS strains that cause neonatal meningitis across different regions and time periods. We applied the PacBio^TM^ SMRT (Pacific Biosciences, Menlo Park, CA, USA) and MiSeq^TM^ (Illumina, San Diego, CA, USA) sequencing technologies for WGS, which have the advantages of easy analysis, few errors, relatively lower cost per sample, and no specific limitations [43,44,45]. However, a gene expression analysis was not performed in this study, and the real mechanisms by which important genes contribute to the occurrence of meningitis were not investigated. Additionally, some early-mortality cases and some invasive GBS strains more than 10 years ago were inevitably lost in this study, which, however, may not have significantly altered the study results.

In conclusion, GBS meningitis in neonates is associated with a higher severity of illness and a high risk of neurological sequelae. GBS meningitis is most likely caused by type III/CC17 and type Ib/CC12 GBS isolates. Genetically, the HvgA, BspC, PezT, and ICE*Sag37* genes, which carries many specific virulence profiles and multiple antibiotic resistance genes, are potentially associated with the enhanced penetrative ability of GBS isolates to cross the BBB and the occurrence of meningitis. Therefore, further in vitro and in vivo models to investigate the roles of these genes on the occurrence of GBS meningitis and the relevance of clinical features are highly suggested. Given such genetic specificity and clinical importance, a multicenter study with more cases of GBS meningitis and GBS isolates is warranted to investigate the molecular mechanisms involved, and the continuous monitoring of neonates with GBS sepsis is critically important in the future.

## 3. Materials and Methods

### 3.1. GBS Isolates, Data Collection, and Definition

We conducted a single-center cohort study that enrolled all neonates with GBS meningitis who were hospitalized in the NICUs of Linkou Chang Gung Memorial Hospital (CGMH) between 2003 and 2020. Linkou CGMH is the largest tertiary-level medical center in Taiwan and located in North Taiwan. There are three NICUs with a total capacity of 47 beds equipped with ventilators and 55 beds in special care nurseries in Linkou CGMH. Meningitis was defined based on the standard criteria [46,47] and all cases had positive GBS strains isolated from cerebrospinal fluid (CSF) cultures. We also analyzed neonates with GBS sepsis but without meningitis during the study period, and this study is part of our longitudinal serial studies. All GBS isolates were retrieved from CGMH’s central laboratory and bacterial library. The clinical information of all the patients, including perinatal demographics, clinical manifestations, laboratory data, hospital courses, and outcomes, were retrospectively reviewed and recorded by our research teams. This study was approved by the Institutional Review Board of CGMH (IRB No. 202102291B0), and a waiver for informed consent for anonymous data collection was approved.

We defined severe sepsis, septic shock, and uncomplicated bacteremia based on the definitions of our previous studies and the Centers for Disease Control and Prevention [23,48]. The presence of neurological complications and long-term neurological sequelae in these patients was evaluated based on the definitions used in the previous studies [14,23]. In addition to EOD and LOD, we categorized late LOD (LLOD), when disease onset was after 90 days of life [49]. The severity of illness was evaluated using the Neonatal Therapeutic Intervention Scoring System (NTISS) score [41] at the onset of meningitis, which was defined at the time of the first positive CSF culture.

### 3.2. Capsular Serotyping, MLST, and Pilus Genes

The capsular serotypes of all GBS isolates were analyzed using the multiplex PCR assay to identify the GBS isolates of types Ia to IX. The DNA isolation method and the PCR assay that amplified and sequenced seven housekeeping genes (*adhP*, *atr*, *glcK*, *glnA*, *pheS*, *sdhA*, and *tkt*) were based on a standard protocol described in our previous publication [10]. Multilocus sequence typing (MLST) was performed based on the standard procedure described in our previous study [42]. After PCR, the sequence type (ST) was assigned based on the allelic profile of each fragment and determined via the *Streptococcus agalactiae* MLST database (http://pubmist.org/sagalactiae, accessed on 1 April 2023). All GBS isolates can be clustered into several major clonal complexes (CCs) based on the goeBURST program [25]. Pilus island content was confirmed via standard multiplex PCR method to identify the pilus island (PI) marker, and a multiplex PCR assay was performed to analyze the distribution of GBS PI genes [26]. We checked all the GBS isolates of neonatal meningitis and sepsis using the target genes of alcohol dehydrogenase gbs0054 (*adhP*) as the housekeeping locus and *sag*647, *sag*1406, and *san*1517 for PI-1, PI-2a, and PI-2b, respectively.

### 3.3. Antimicrobial Susceptibility Testing

Antimicrobial susceptibility testing was performed for all the GBS isolates using the disc diffusion method as described in previous studies [27]. The double-disk diffusion test was applied to identify inducible clindamycin resistance. All GBS isolates were tested for susceptibility to seven antibiotics, including erythromycin, penicillin, clindamycin, vancomycin, ampicillin, cefotaxime, and teicoplanin, according to the guidelines of Clinical and Laboratory Standards Institute (CLSI) for the disc diffusion method [28].

### 3.4. Whole Genome Sequencing

Three GBS isolates selected from the III/ST-17 (termed N48, N96, and N5) strains, one type Ib/ST-12 (termed N92), and three type VI/ST-1 (termed N55, N70, and N132) GBS strains were used for whole-genome sequencing (WGS). The type III/ST-17 and type Ib/ST-12 strains were obtained from the cerebrospinal fluid of neonates with meningitis, while the type VI/ST-1 GBS isolates were from the blood culture of neonates with sepsis without meningitis. The GBS isolates selected to perform the WGS were based on previous experiences and that we have found similar WGS results among the same GBS serotypes and sequence types. Additionally, type III/ST-17 and Ib/ST-12 GBS isolates were among the most common GBS strain to cause meningitis. The standard protocol of the lysozyme-sodium dodecyl sulfate-proteinase K method was used to extract DNA. WGS was performed using both PacBio^TM^ SMRT (Pacific Biosciences, Menlo Park, CA, USA) [43] and MiSeq^TM^ (Illumina, San Diego, CA, USA) [44] sequencing technologies. The sequencing library was prepared using a TruSeq DNA LT Sample Prep Kit (Illumina, San Diego, CA, USA) for the Illumina MiSeq system. Genomic libraries were generated using Nextera XT kits (Illumina, San Diego, CA, USA). We used SPAdes (version 3.9.0) to assemble the sequences. All genome sequences were subjected to BLAST analysis using the NCBI genome database to identify possible plasmid sequences. After the de novo-assembled genome was generated, Prokka (v1,12) [50] was used for the genome annotation and identification of rRNA-encoding and tRNA-encoding regions.

### 3.5. Statistical Analysis

The clinical and genetic characteristics were compared between neonates with GBS meningitis and those with GBS sepsis but without meningitis during the study period. Categorical and continuous variables are expressed as proportions and the median (interquartile, IQR), respectively. Categorical variables were compared using the χ^2^ test or Fisher’s exact test; odds ratios (ORs) and 95% confidence intervals (CIs) were calculated. Continuous variables were compared using the Mann–Whitney *U* test and the *t*-test, depending on the distribution. The trend of the proportions of categorical variables among the subgroups was analyzed via the Cochran–Armitage trend test. The results with *p* values of <0.05 were considered statistically significant. All statistical analyses were performed using SPSS version 23 (IBM SPSS Statistics).

## Figures and Tables

**Figure 1 ijms-24-15387-f001:**
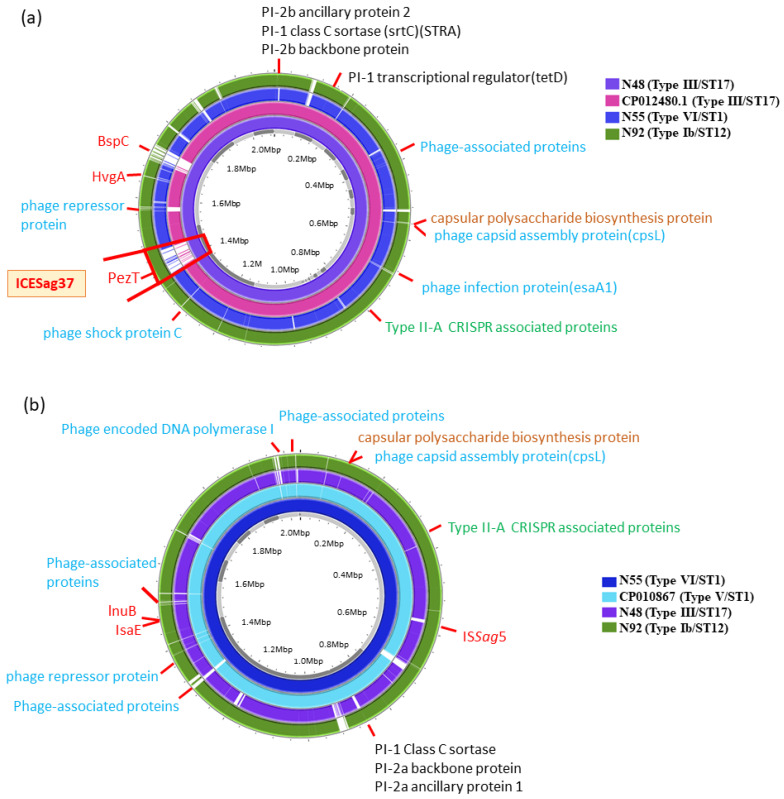
Whole-genome sequence analysis of one type III/ST17 GBS strain (N48), one Ib/ST12 GBS strain (N92), one VI/ST1 GBS strain (N55), and two reference strains, CP012480.1 (type III/ST17) and CP010867 (type V/ST1). The genome scales in mega base pairs of these two reference strains are given in the inner circle of (**a**,**b**), respectively. The N48 and N92 are from neonates with GBS meningitis, and N55 represents neonatal sepsis without meningitis. TBLASTN comparisons of the genomes of the reference GBS strains are shown in different colors and compared with the complete genomes of three clinical GBS isolates. Of note, the HspC, PezT, and ICE*Sag37* genes are present in N48 and N92, but absent in N55. The HvgA gene is type III/ST-17 GBS-specific and noted in N48 only. Additionally, several genes encoding several phage-associated proteins, PI-1- and PI-2a-associated proteins, and type I CRISPR-associated proteins are present in type Ib/ST12 GBS strains only.

**Table 1 ijms-24-15387-t001:** Patient demographics and clinical features of neonates with Group B *Streptococcus* (GBS) meningitis and those with GBS sepsis but without meningitis from Chang Gung Memorial Hospital (CGMH), 2003–2020.

	Neonates with GBS Meningitis (Total n = 48)	Neonates with GBS Sepsis without Meningitis (n = 140)	*p* Values
Gestational age, (week)	38.0 (37.0–39.0)	38.0 (36.0–39.8)	0.956
Birth body weight, (g)	2900.0 (2651.3–3251.3)	2885.0 (2412–3236.3)	0.481
Gender, (male/female, n/%)	19 (39.6)/29 (60.4)	62 (44.3)/78 (55.7)	0.615
Birth via NSD/Cesarean section, n (%)	32 (66.7)/16 (33.3)	98 (70.0)/42 (30.0)	0.718
5 min Apgar score < 7, n (%)	0 (0)	12 (8.6)	0.036
Premature rupture of membrane, n (%)	7 (14.6)	28 (20.0)	0.521
Onset of GBS bacteremia (day), median (IQR)	19.5 (7.3–33.8)	30.0 (11.0–54.8)	0.229
Early-onset sepsis (≤7 days), n (%)	12 (25.0)	32 (22.9)	0.826
Late-onset sepsis (8–90 days), n (%)	34 (70.8)	99 (70.7)	
Very-late onset sepsis (>90 days), n (%)	2 (4.2)	9 (6.4)	
Clinical features *, n (%)			
Fever (≥38.3 °C)	43 (89.6)	110 (78.6)	0.131
Apnea, bradycardia, and/or cyanosis	27 (56.3)	45 (32.1)	0.005
Ventilator requirement			0.001
Room air	21 (43.8)	95 (67.9)	
Nasal canula	3 (6.3)	5 (3.6)	
Non-invasive ventilator (N-CPAP and N-IMV)	3 (6.3)	15 (10.7)	
Intubation	20 (41.7)	17 (12.1)	
High-frequency oscillatory ventilator	1 (2.1)	8 (5.7)	
Abdominal distension and/or vomiting	27 (56.3)	44 (31.4)	0.003
Hypoglycemia	5 (10.4)	17 (12.1)	1.000
Hypotension	12 (25.0)	22 (15.7)	0.191
Severe sepsis	22 (45.8)	39 (27.9)	0.031
Disseminated intravascular coagulopathy	8 (16.7)	6 (4.3)	0.009
Requirement of blood transfusion **	28 (58.3)	60 (42.9)	0.068
Laboratory data at onset of GBS bacteremia, n (%)			
Leukocytosis (WBC > 20,000/L)	25 (52.1)	85 (60.7)	0.313
Leukopenia (WBC < 4000/L)	17 (35.4)	24 (17.1)	0.014
Shift to left in WBC (immature > 20%)	10 (20.8)	13 (9.3)	0.043
Anemia (hemoglobin level < 11.5 g/dL)	27 (56.3)	70 (50.0)	0.505
Thrombocytopenia (platelet < 150,000/μL)	11 (22.9)	19 (13.6)	0.169
Metabolic acidosis	12 (25.0)	15 (10.7)	0.029
Coagulopathy	13 (27.1)	17 (12.1)	0.022
C-reactive protein (mg/dL), median (IQR)	123.8 (45.8–187.9)	21.8 (8.3–57.7)	<0.001

* At onset of bacterial bacteremia. ** Including leukocyte poor red blood cell and/or platelet transfusion. All data are expressed as numbers (%) or medians (IQR). IQR: interquartile range; WBC: white blood cell count; N-CPAP: nasal continuous positive airway pressure; N-IMV: non-invasive mechanical ventilation.

**Table 2 ijms-24-15387-t002:** Neurological complications in neonates with Group B streptococcal (GBS) meningitis and sepsis without meningitis in CGMH, 2003–2020.

Neurological Complications, Sequelae and Death	Neonates with GBS Meningitis (n = 48)	Neonates with GBS Sepsis without Meningitis (n = 140)
Any neurological complications	37 (77.8)	8 (5.7)
Seizure	22 (45.8)	5 (3.6)
Subdural effusion	16 (33.3)	1 (0.7)
Increased intracranial pressure	12 (25.0)	7 (5.0)
Ventriculomegaly	17 (35.4)	0 (0)
Hydrocephalus	6 (12.5)	1 (0.7)
Encephalomalacia	6 (12.5)	0 (0)
Subependymal hemorrhage	5 (10.4)	2 (1.4)
Intraventricular hemorrhage	4 (8.3)	4 (2.9)
Ventriculitis	4 (8.3)	0 (0)
Periventricular leukomalacia	1 (2.1)	1 (0.7)
Infarction	5 (10.4)	0 (0)
Subdural empyema or abscess	2 (4.2)	0 (0)
Brain atrophy	1 (2.1)	0 (0)
Discharge with neurological sequelae	17 (41.5)	4 (2.9)
Final in-hospital mortality	7 (14.6)	12 (8.6)

All data are expressed as numbers (%).

**Table 3 ijms-24-15387-t003:** The genetic characteristics, sequence types, clonal complexes, and serotypes of all *Streptococcus agalactiae* (GBS) isolates causing neonatal meningitis and sepsis in CGMH, 2003–2020.

	GBS Isolates of Neonatal Meningitis (Total n = 48)		GBS Isolates of Neonatal Sepsis without Meningitis (Total n = 140)	
Serotypes	Type III GBS strains (n = 33)	Non-Type III GBS strains (n = 15)	Serotype III GBS strains (n = 92)	Non-Type III GBS strains (n = 48)
Sequence types	ST17 (27), ST19 (5), ST438 (1)	ST1 (1), ST12 (4), ST23 (5), ST24 (3), ST268 (2)	ST17 (85), ST19 (5), ST335 (1), ST890 (1)	ST1 (14), ST10 (1), ST12 (10), ST23 (11), ST24 (6), ST144 (1), ST452 (1), ST890 (4)
Clonal Complex	CC17 (27), CC19 (5), CC438 (1)	CC1 (1), CC12 (6), CC23 (5), CC24 (3)	CC17 (85), CC19 (6), CC890 (1)	CC1 (14), CC12 (11), CC23 (12), CC24 (6), CC144 (1), CC890 (4)
Genetic characteristics				
Pilus genes				
PI-1 + PI-2a	4 (12.1)	5 (33.3)	6 (6.5)	24 (50.0)
PI-1 + PI-2b	8 (24.2)	0 (0)	9 (9.8)	1 (2.1)
PI-2a only	0 (0)	10 (66.7)	1 (1.1)	21 (43.8)
PI-2b only	21 (63.6)	0 (0)	76 (82.6)	2 (4.2)
Virulence genes				
PezT	20 (60.6)	9 (60.0)	76 (82.6)	21 (43.8)
HvgA	27 (81.8)	0 (0)	85 (92.4)	0 (0)
BspC	28 (84.8)	9 (60.0)	79 (85.9)	26 (54.2)
ICE*Sag37*	20 (60.6)	5 (33.3)	76 (82.6)	12 (25.0)
IS*Sag5*	8 (24.2)	2 (13.3)	10 (10.9)	13 (27.1)
Antibiotic resistance genes				
lsa(E)	5 (15.2)	7 (46.7)	22 (23.9)	14 (29.2)
lun(B)	5 (15.2)	7 (46.7)	22 (23.9)	14 (29.2)
Antibiotic resistance patterns [No. (%) of resistant GBS isolates]				
Ery (R) + Clin (R)	23 (69.7)	6 (40.0)	79 (85.9)	21 (43.8)
Ery (S) + Clin (S)	8 (24.2)	8 (53.3)	9 (9.8)	26 (54.2)
Ery (R) + Clin (S)	2 (6.1)	0 (0)	3 (3.3)	1 (2.1)
Ery (S) + Clin (R)	0 (0)	1 (6.7)	1 (1.1)	0 (0)

Ery: erythromycin; Clin: clindamycin; R: resistant; S: susceptible. All GBS isolates are susceptible to vancomycin, teicoplanin, ampicillin, penicillin, and cefotaxime.

**Table 4 ijms-24-15387-t004:** All primers used for targeted genes in PCR and the relative positions of these genes inside or around the integrative and conjugative element, ICE*Sag37*, pilus genes, and various virulence genes.

Gene	Sequence (5′-> 3′ Y)	Product Size (bp)
HvgA	F: ATGTTTACGAAAAAGTTAAACCAG	204
	R: CCAAGTTTCCGCTAGTATTAACCG	
BspC	F: ATATTTTGAGGGCAAGATCGC	376
	R: AGGTCCAGCTTCAAATCCTTC	
ICE*Sag37* head	F: ACATAGCCCCGTCAGTATG	816
	R: ATCACGTGGAGTGGTAGTG	
ICE*Sag37* tail	F: GCAACGTGGTGAATTGATAGGG	1011
	R: AAAACTGCACGATCAAACTCCG	
lea(E)	F: TGTCAAATGGTGAGCAAACG	495
	R: TGTAAAACGGCTTCCTGATG	
Inu(B)	F: ACCAAAGGAGAAGGTGACCAA	584
	R: ACCTTATCTAATCGAGCAGTGGT	
PezT	F: ATACGAAAATTTACCTTGTCGC	926
	R: TAAATCCTCGCAATTCTAACCC	
PI-1	F: CAAGATTGACCGGGTGGAGA	325
	R: ATGGGCAGTTAGAACGGCAT	
PI-2a	F: CGGGGTGCAAGTCAATAAGG	264
	R: GGAGCAGGGCATTTAGAAGGT	
PI-2b	F: CTCTGCTACCACCAAAGCGT	665
	R: GTGGGGGTAGGCTTAATGGC	

## Data Availability

The datasets used/or analyzed during the current study are available from the corresponding author on reasonable request.

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
