# Peer review of "The Clinical and Genetic Characteristics of Streptococcus agalactiae Meningitis in Neonates"

_ijms, 2023, doi:10.3390/ijms242015387_

Round 1

Reviewer 1 Report

Overall, your manuscript presents a comprehensive exploration of the molecular characteristics and antimicrobial resistance patterns of Group B Streptococcus (GBS) isolates associated with neonatal meningitis. The combination of clinical data, epidemiological analysis, and genetic sequencing provides valuable insights into the factors contributing to GBS meningitis in neonates. While I find your work to be of high quality and significance, I have outlined some specific comments and suggestions below that I believe would further enhance the manuscript:

- Title and Abstract: The title accurately reflect the content of the study. However, I suggest revisiting the abstract to emphasize the principal findings more explicitly for readers seeking a concise overview of the research.

- Study Population: While the manuscript describes the study population well, it would be beneficial to specify the geographic location of the study population. Additionally, clarify whether the neonates were sourced from a single medical center or multiple centers.

- Genomic Analysis: Offer more details regarding the sequencing methods employed, such as the sequencing platforms (e.g., Illumina, PacBio) and the specific bioinformatics tools used. Elaborating on the rationale for selecting these methods and discussing their limitations would enhance the reproducibility of your research.

- The discussion section effectively links your findings to previous research. However, it would be valuable to delve deeper into the clinical implications of your results. Also, discuss potential limitations of your study, such as potential biases or challenges encountered during data collection or analysis. Suggest areas for future research or experiments that could build upon your findings and address existing knowledge gaps.

- Your conclusion is well-structured and summarizes the key findings effectively. Reiterate the clinical significance of the results briefly in the conclusion to underscore their importance to the field.

- I appreciate the comprehensive research presented in your manuscript. I wonder if you have considered investigating the virulence factors associated with GBS isolates causing neonatal meningitis. Understanding the specific factors that contribute to the enhanced pathogenicity of certain strains could potentially have significant clinical implications. Could you kindly share your thoughts on the possibility of conducting such investigations or any insights you might have regarding this aspect?".

- Explore whether specific genetic mutations or mechanisms contribute to resistance and if these can be targeted for intervention.

-Personalized Preventive Strategies: Exploring host factors could potentially lead to personalized preventive strategies. If we can identify specific genetic markers or immune profiles that increase susceptibility, it might pave the way for tailored interventions for at-risk neonates. How do you see this prospect impacting clinical practice in the long term?

 - Enhanced Understanding: A comprehensive understanding of neonatal GBS meningitis requires us to consider both bacterial virulence factors and host-related factors. By conducting such investigations, we may gain insights into the interplay between bacterial pathogens and the host's defense mechanisms. How might this contribute to a more holistic understanding of the disease?

-Investigate host factors that may predispose neonates to GBS meningitis. Explore whether there are genetic or immunological factors that increase susceptibility, potentially leading to personalized preventive strategies.

-Examine the potential role of maternal colonization with GBS in neonatal infection. A study on the dynamics of GBS transmission from mother to neonate and strategies to reduce vertical transmission could be valuable.

 By addressing the specific comments, suggestions, and the supportive experiments or studies outlined above, you can further elevate the quality and impact of your work. I encourage you to consider these recommendations as you revise and expand upon your manuscript. I look forward to seeing your revised submission.

Thank you for your contribution to scientific research, and I wish you every success with your future work.

Language and Clarity: Although your manuscript is generally well-written and clear, there are some complex sentences that could be simplified for improved readability. Please also review the manuscript for any typographical errors or grammatical issues.

Author Response

Dear reviewer:

I appreciate your review and comments. Please see the attached file, thank you.

Best regard,

Tsai Ming Horng

Reviewer 2 Report

In the here presented manuscript entitled "The clinical and genetic characteristics of Streptococcus agalactiae meningitis in neonates", Hsu et al. set out to investigate clinical features and molecular characteristics of Group B Streptococcus (GBS) isolates causing neonatal meningitis. This is an interesting retrospective study. The authors compare these isolates to cases of sepsis without meningitis. The molecular techniques used for characterization of the GBS isolates, including serotyping, MLST, antimicrobial susceptibility testing, and whole genome sequencing are sound. The results highlight the significantly worse clinical symptoms associated with GBS meningitis in neonates and provide some interesting genomic clues about GBS invasiveness using whole genome sequencing. The discussion interprets the main findings in the context of the current literature and hypothesizes about the potential role of genes, including BspC, PezT, and ICESag37 in GBS pathogenesis and meningitis. The limitations of the study are acknowledged.

Specific points:

1.     Explain the rationale for selecting the specific meningitis and sepsis GBS isolates for whole genome sequencing. The authors did not specify what criteria were used in lines 109-111 or 196-198.

2.     In the results section, it would be helpful to briefly explain the genes BspC, PezT, and HvgA when first mentioned for readers who may be unfamiliar with them.  

3.     In the discussion, elaborate on the possible mechanisms for the identified genes in promoting blood-brain barrier penetration and meningeal invasion, specifically in neonates. For example, discuss specific host factors that might be involved in the GBS invasiveness in <7 day-old neonates.

4.     Were there any differences in maternal risk factors or intrapartum antibiotic prophylaxis between the meningitis and sepsis groups? This data could provide further insights.

Overall, this study provides useful genomic and clinical insights into the pathogenesis of neonatal GBS meningitis. Addressing the above minor points would further strengthen the manuscript. The topic is of significant interest and importance.

Overall, the manuscript is proficiently written in scientific English with only minor editing needed, and the language used is suitable.

Suggestions for improvement:

1.    Some long sentences could be shortened or split for easier readability.

2.    Minor rewording for improved flow and clarity can be made in lines 302-306: All the invasive GBS isolates in this study were from a single center in Taiwan over an extensive period of time. During this prolonged timeframe, the therapeutic and IAP policies may have been changed. Therefore, a multicenter study is needed to investigate the epidemiological and genetic trends of GBS strains that cause neonatal meningitis across different regions and time periods.

Author Response

(The authors gave the same response as above.)

Round 2

Reviewer 1 Report

The manuscript has been revised with attention to the reviewers' comments and can now be published.